# Flavonoid intake is associated with lower mortality in the Danish Diet Cancer and Health Cohort

Nicola P. Bondonno [1,2,12], Frederik Dalgaard [3,12], Cecilie Kyrø [4], Kevin Murray [5], Catherine P. Bondonno [1,2], Joshua R. Lewis [1,2], Kevin D. Croft [1], Gunnar Gislason[3,6,7], Augustin Scalbert [8], Aedin Cassidy [9], Anne Tjønneland [4], Kim Overvad [10,11] & Jonathan M. Hodgson [1,2]

Flavonoids, plant-derived polyphenolic compounds, have been linked with health benefits. However, evidence from observational studies is incomplete; studies on cancer mortality are scarce and moderating effects of lifestyle risk factors for early mortality are unknown. In this prospective cohort study including 56,048 participants of the Danish Diet, Cancer, and Health cohort crosslinked with Danish nationwide registries and followed for 23 years, there are 14,083 deaths. A moderate habitual intake of flavonoids is inversely associated with all-cause, cardiovascular- and cancer-related mortality. This strong association plateaus at intakes of approximately 500 mg/day. Furthermore, the inverse associations between total flavonoid intake and mortality outcomes are stronger and more linear in smokers than in non-smokers, as well as in heavy (>20 g/d) vs. low-moderate (<20 g/d) alcohol consumers. These findings highlight the potential to reduce mortality through recommendations to increase intakes of flavonoid-rich foods, particularly in smokers and high alcohol consumers.

[1] School of Medical and Health Sciences, Edith Cowan University, Perth, Western Australia, Australia. [2] School of Biomedical Sciences, University of Western Australia, Royal Perth Hospital, Perth, Western Australia, Australia. [3] Department of Cardiology, Herlev & Gentofte University Hospital, Copenhagen, Denmark. [4] The Danish Cancer Society Research Centre, Copenhagen, Denmark. [5] School of Population and Global Health, University of Western Australia, Crawley, Western Australia, Australia. [6] The National Institute of Public Health, University of Southern Denmark, Odense, Denmark. [7] The Danish Heart Foundation, Copenhagen, Denmark. [8] International Agency for Research on Cancer, Lyon, France. [9] Institute for Global Food Security, Queen's University Belfast, Belfast, Northern Ireland. [10] Department of Public Health, Aarhus University, Aarhus, Denmark. [11] Aalborg University Hospital, Aalborg, Denmark. [12]These authors contributed equally: Nicola P. Bondonno, Frederik Dalgaard. Correspondence and requests for materials should be addressed to N.P.B. (email: n.bondonno@ecu.edu.au)

Fruit and vegetable intakes are associated with a lower risk of cardiovascular disease (CVD), cancer, and all-cause mortality, with an estimated 7.8 million premature deaths worldwide in 2013 attributable to a fruit and vegetable intake below 800 g/day[1]. These benefits are likely due in part to flavonoids, a class of polyphenolic compounds found in abundance in plant-derived foods and beverages such as fruits, vegetables, dark chocolate, tea, and red wine[2]. Flavonoids are categorized into six main subclasses based on their chemical structure: flavonols, flavan-3-ols, flavanones, flavones, anthocyanins, and isoflavones. Structural differences bring about variations in metabolism and bioactivity[3], which may result in varying effects on health outcomes[4].

Evidence for the potential health benefits of flavonoids comes from epidemiological studies, short-term randomized controlled trials, and preclinical studies. Short-term clinical trials have demonstrated that flavonoid compounds and flavonoid-rich foods improve surrogate measurements of CVD risk[5] Results from preclinical studies suggest that flavonoids may also modulate the risk of cancer[6]. Evidence from observational studies is incomplete; studies on cancer mortality are scarce and additional research is necessary to establish the specific role of flavonoid subclasses and to determine the dose of total and specific flavonoids required to achieve maximum benefit[7]. Additionally, emerging evidence suggests that flavonoids may afford greater protection to those with lifestyle habits placing them at risk of early mortality[8]. Flavonoids may protect against some of the detrimental effects that these factors have on nitric oxide bioavailability, endothelial function, blood pressure, inflammation, blood lipids, platelet function, and/or thrombosis[9].

The primary aim of this study was to investigate the association of total flavonoid and flavonoid subclass intakes with all-cause, CVD-related, and cancer-related mortality in 56,048 participants of the Danish Diet, Cancer, and Health cohort. We demonstrate that an achievable dietary intake of total and individual flavonoid subclasses is associated with a lower risk of all-cause, CVD-related, and cancer-related mortality. Secondary aims were to investigate whether these associations differed according to the presence of modifiable lifestyle risk factors. We demonstrate that inverse associations are strongest for current smokers and individuals with high alcohol consumption.

## Results

**Baseline characteristics**. In this population of 56,048 Danish citizens, with a median [IQR] age of 56 [52–60] years at entry, 14,083 died from any cause during 1,085,186 person-years of follow-up. Of the 52,492 participants without CVD at baseline, 4065 died of CVD and of the 55,801 participants without cancer at baseline, 6299 died of cancer. The baseline characteristics of the study population overall and stratified by total flavonoid intake quintiles are shown in Table 1. The median [IQR] total flavonoid intake was 494 [286–804] mg/d and the distribution was skewed-right. Pearson's correlations between flavonoid subclass intakes ranged from 0.97 for the flavonols and flavanol monomers to 0.01 for the flavanol monomers and the flavanones (Supplementary Table 1). Compared to participants in the lowest quintile of total flavonoid intake, those in the highest quintile tended to have a lower BMI and be more physically active, and were less likely to be current smokers and more likely to be female, have a higher level of education, and a higher income. They also tended to have a lower prevalence of heart failure, ischemic heart disease, stroke, and COPD. Participants with higher flavonoid intake consumed more dietary fiber and less red meat and processed meat.

**Total and flavonoid subclasses and all-cause mortality**. For total flavonoids and all flavonoid subclasses, the test of nonlinearity was statistically significant (P-nonlinearity < 0.001). Evidence of a threshold effect for the association between total flavonoid intake and all-cause mortality can be seen in Fig. 1, where the inverse relationship plateaued beyond intakes of approximately 500 mg/d. A similar association was seen for all flavonoid subclasses, with the threshold occurring at varying levels of intake, except the flavone and flavanone subclasses, which were slightly U-shaped (Fig. 2). HRs for quintiles of total flavonoid and flavonoid subclass intakes are shown in Table 2. For total flavonoid intake, after multivariable adjustments (model 2), participants in the second and third quintile were at a lower risk of all-cause mortality [HR (95% CI), Q2: 0.88 (0.85, 0.91) and Q3:0.83 (0.80, 0.86)]. At higher levels of the exposure, the HR remained constant. In general, adjustments for potential lifestyle and dietary confounders (model 3) slightly attenuated the association and HRs for all flavonoid subclasses were not substantially lower beyond quintile 3.

**Total and flavonoid subclasses and cause-specific mortality**. Intakes above those in quintile 1 for total flavonoids and all flavonoid subclasses were associated with a lower risk of both CVD- and cancer-related mortality after adjustment for potential lifestyle and dietary confounders. For CVD-related mortality, the association plateaued beyond intakes of approximately 500 mg/d of total flavonoids (Fig. 1) and in general, the associations were stable beyond quintile 3 for any subclass (Table 3 and Supplementary Fig. 1). For cancer-related mortality, the association plateaued at approximately 1000 mg/d of total flavonoids (Fig. 1) and for the flavonol, flavanol oligo + polymer subclasses, lower HRs were observed until quintile 5 (Table 4 and Supplementary Fig. 2).

**Stratified analyses**. The associations between flavonoid intake and both all-cause and cause-specific mortality differed according to smoking status, alcohol intake, and BMI, but not by sex, level of physical activity, or presence of prevalent diabetes.

As there was no difference in association between flavonoid intake and mortality for never vs. former smokers, these were collapsed into one subgroup. The inverse association between total flavonoid intake and both all-cause and cause-specific mortality was stronger in smokers than in non-smokers as well as in those who consumed >20 g/d of alcohol compared to those who consumed ≤20 g/d of alcohol (Fig. 3 and Supplementary Figs. 3 and 4). These benefits appeared extend to higher doses of up to 1000–2000 mg/d. Evidence of a dose–response association can be seen in Supplementary Fig. 5, where the association between total flavonoid intake and both all-cause and cancer-related mortality tended to stronger at higher smoking intensity (assessed by pack-years) and alcohol intake (g/d). Similar, although less clear, interactions between flavonoid intake and both smoking intensity and alcohol intake were seen for CVD-related mortality.

In general, the lower risk of mortality associated with higher flavonoid intake tended to be weaker in obese (BMI > 30) participants in comparison to normal/overweight (18.5 > BMI ≤ 30) participants (Fig. 3 and Supplementary Fig. 5).

**Sensitivity analyses**. Neither including energy intake in model 2, nor removing hypertension, hypercholesterolemia, and prevalent diseases from model 2, or adjusting for medication use substantively altered the hazard ratios (Supplementary Table 2). Excluding all participants with comorbidities at baseline (n = 5 492) marginally strengthened the relationship of total and individual flavonoid subclass intake with all-cause mortality

**Table 1 Baseline characteristics of study population**

| | Total population n = 56 048 | Total flavonoid intake quintiles | | | | |
|---|---|---|---|---|---|---|
| | | 1 n = 11 210 | 2 n = 11 209 | 3 n = 11 210 | 4 n = 11 209 | 5 n = 11 210 |
| Total flavonoid intake (g/d) | 494 [286–804] | 173 [127–212] | 320 [286–356] | 494 [441–548] | 726 [659–804] | 1201 [1024–1435] |
| Sex (male) | 26666 (47.6) | 6477 (57.8) | 5740 (51.2) | 5340 (47.6) | 4996 (44.6) | 4113 (36.7) |
| Age (years) | 56 [52–60] | 56 [52–60] | 56 [52–60] | 56 [52–60] | 56 [52–60] | 55 [52–60] |
| BMI (kg/m$^2$) | 25.5 [23.3–28.2] | 26.1 [23.8–28.9] | 25.9 [23.6–28.5] | 25.6 [23.3–28.3] | 25.3 [23.2–27.9] | 24.9 [22.7–27.4] |
| MET score | 56 [37–85] | 51 [32–78] | 56 [36–84] | 58 [38–85] | 58 [39–87] | 60 [40–89] |
| Smoking status | | | | | | |
| Never | 19666 (35.1) | 2722 (24.3) | 3743 (33.4) | 3996 (35.6) | 4461 (39.8) | 4744 (42.3) |
| Former | 16153 (28.8) | 2695 (24.0) | 3033 (27.1) | 3266 (29.1) | 3597 (32.1) | 3562 (31.8) |
| Current | 20229 (36.1) | 5793 (51.7) | 4433 (39.5) | 3948 (35.2) | 3151 (28.1) | 2904 (25.9) |
| Education: ≤7 years | 18466 (32.9) | 5128 (45.7) | 4268 (38.1) | 3609 (32.2) | 3031 (27.0) | 2430 (21.7) |
| 8–10 years | 25817 (46.1) | 4891 (43.6) | 5263 (47.0) | 5341 (47.6) | 5317 (47.4) | 5005 (44.6) |
| ≥11 years | 11737 (20.9) | 1185 (10.6) | 1674 (14.9) | 2256 (20.1) | 2854 (25.5) | 3768 (33.6) |
| Household income (DKK/year) | | | | | | |
| ≤394 700 | 13919 (24.8) | 3349 (29.9) | 2749 (24.5) | 2718 (24.2) | 2581 (23.0) | 2522 (22.5) |
| 394 701–570 930 | 14018 (25.0) | 3271 (29.2) | 3003 (26.8) | 2717 (24.2) | 2603 (23.2) | 2424 (21.6) |
| 570 931–758 297 | 14054 (25.1) | 2916 (26.0) | 3034 (27.1) | 2900 (25.9) | 2621 (23.4) | 2583 (23.0) |
| >758 297 | 14057 (25.1) | 1674 (14.9) | 2423 (21.6) | 2875 (25.6) | 3404 (30.4) | 3681 (32.8) |
| Hypertensive | 9148 (16.3) | 1815 (16.2) | 1860 (16.6) | 1866 (16.6) | 1823 (16.3) | 1784 (15.9) |
| Hypercholesterolemic | 4193 (7.5) | 915 (8.2) | 831 (7.4) | 853 (7.6) | 857 (7.6) | 737 (6.6) |
| Diabetes | 1182 (2.1) | 279 (2.5) | 221 (2.0) | 254 (2.3) | 217 (1.9) | 211 (1.9) |
| Heart failure | 220 (0.4) | 56 (0.5) | 54 (0.5) | 39 (0.3) | 40 (0.4) | 31 0.3) |
| Atrial fibrillation | 451 (0.8) | 95 (0.8) | 83 (0.7) | 96 (0.9) | 78 (0.7) | 99 (0.9) |
| IHD | 2200 (3.9) | 586 (5.2) | 423 (3.8) | 438 (3.9) | 400 (3.6) | 353 (3.1) |
| PAD | 498 (0.9) | 171 (1.5) | 114 (1.0) | 84 (0.7) | 61 (0.5) | 68 (0.6) |
| Stroke | 787 (1.4) | 220 (2.0) | 151 (1.3) | 147 (1.3) | 133 (1.2) | 136 (1.2) |
| COPD | 858 (1.5) | 223 (2.0) | 189 (1.7) | 157 (1.4) | 155 (1.4) | 134 (1.2) |
| CKD | 204 (0.4) | 43 (0.4) | 33 (0.3) | 44 (0.4) | 42 (0.4) | 42 (0.4) |
| Cancer | 247 (0.4) | 55 (0.5) | 42 (0.4) | 61 0.5) | 33 (0.3) | 56 (0.5) |
| Insulin treated | 381 (0.7) | 78 (0.7) | 65 (0.6) | 86 (0.8) | 81 (0.7) | 71 (0.6) |
| Antihypertensive | 6904 (12.3) | 1363 (12.2) | 1422 (12.7) | 1407 (12.6) | 1363 (12.2) | 1349 (12.0) |
| Statin | 1074 (1.9) | 262 (2.3) | 211 (1.9) | 221 (2.0) | 212 (1.9) | 168 (1.5) |
| HRT | | | | | | |
| Never | 15972 (54.4) | 2606 (55.1) | 3049 (55.8) | 3268 (55.7) | 3271 (52.6) | 3778 (53.2) |
| Current | 8825 (30.0) | 1295 (27.4) | 1572 (28.7) | 1698 (28.9) | 2001 (32.2) | 2259 (31.8) |
| Former | 4553 (15.5) | 823 (17.4) | 844 (15.4) | 897 (15.3) | 935 (15.0) | 1054 (14.9) |
| NSAIDs excluding aspirin | 18161 (32.6) | 3543 (31.9) | 3530 (31.7) | 3642 (32.7) | 3640 (32.6) | 3806 (34.2) |
| Aspirin | 7097 (12.7) | 1392 (12.4) | 1366 (12.2) | 1437 (12.8) | 1395 (12.4) | 1507 (13.4) |
| Energy (kj) | 9500 [7858–11368] | 8610 [7026–10387] | 9262 [7717–11000] | 9752 [8139–11586] | 9938 [8321–11834] | 9928 [8258–11887] |
| Saturated FA (g/d) | 31 [24–40] | 29 [23–38] | 31 [24–39] | 32 [25–40] | 32 [25–41] | 32 [24–41] |
| Polyunsaturated FA (g/d) | 13 [10–17] | 12 [9–16] | 13 [10–17] | 14 [10–18] | 14 [11–18] | 14 [10–18] |
| Monounsaturated FA (g/d) | 27 [21–35] | 26 [20–34] | 27 [21–35] | 28 [22–35] | 28 [22–35] | 27 [21–34] |
| Total fish intake (g/d) | 38 [25–55] | 33 [22–49] | 38 [25–54] | 40 [27–57] | 41 [28–59] | 40 [27–57] |
| Red meat intake (g/d) | 78 [57–107] | 80 [58–108] | 81 [59–110] | 80 [58–110] | 78 [57–107] | 72 [52–100] |
| Processed meat intake (g/d) | 25 [14–40] | 29 [17–45] | 26 [16–42] | 25 [14–40] | 23 [14–38] | 21 [12–34] |
| Dietary fiber intake (g/d) | 20 [16–25] | 17 [13–20] | 19 [16–23] | 21 [17–25] | 22 [18–27] | 23 [19–29] |
| Vegetable intake (g/d) | 161 [105–231] | 114 [71–170] | 150 [99–211] | 168 [113–235] | 184 [127–254] | 196 [135–272] |
| Fruit intake (g/d) | 171 [95–281] | 87 [44–140] | 161 [97–237] | 193 [114–301] | 224 [139–360] | 24 [141–390] |
| Alcohol intake (g/d) | 13 [6–31] | 11 [3–24] | 13 [6–25] | 15 [6–34] | 14 [7–32] | 13 [6–32] |

Data expressed as median [IQR] or n (%) unless otherwise stated
*BMI* body mass index, *CKD* chronic kidney disease, *COPD* common obstructive pulmonary disease, *CVD* cardiovascular disease, *DKK* Danish Krone, *FA* fatty acids, *HRT* hormone replacement therapy, *IHD* ischemic heart disease, *MET* metabolic equivalent, *NSAID* Nonsteroidal anti-inflammatory drug, *PAD* peripheral artery disease

(Supplementary Table 2). When stratifying by tertiles of total fruit and vegetable intake, evidence of an association between total flavonoid intake and all-cause mortality remained amongst participants in the highest intake tertile (Supplementary Table 3). There was no association between total flavonoid intake and our falsification endpoint; a burn or foreign object [n = 3 020; HR (95%CI) for Q5 vs Q1: 1.01 (0.92, 1.12); Supplementary Table 4].

## Discussion

There is great potential to improve population health through improved dietary behaviors, with tailored recommendations focusing on specific dietary components and population sub-groups. In this prospective cohort study of 56,048 Danes, we provide evidence that an achievable dietary intake of total and individual flavonoid subclasses is associated with a lower risk of all-cause, CVD-related, and cancer-related mortality. Our results

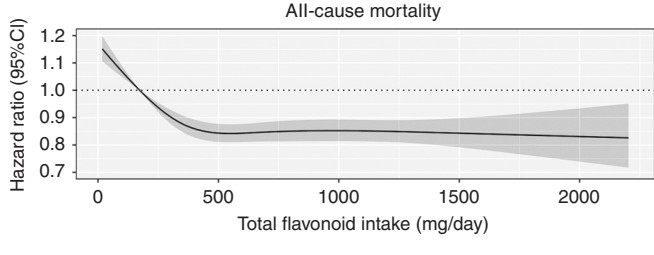

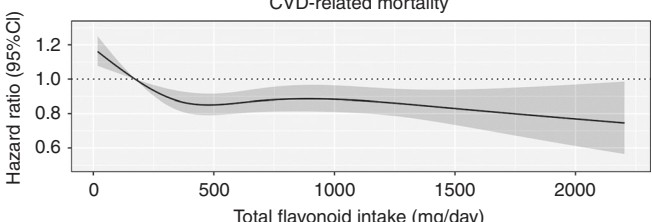

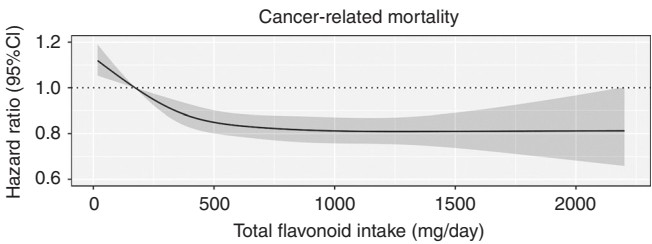

**Fig. 1** The association of total flavonoid intake with all-cause and cause-specific mortality. Hazard ratios are based on Cox proportional hazards models adjusted for age, sex, BMI, smoking status, physical activity, alcohol intake, hypertension, hypercholesterolemia, social economic status (income), and prevalent disease and are comparing the specific level of flavonoid intake (horizontal axis) to the median intake for participants in the lowest intake quintile

indicate a threshold, with no added benefit observed beyond 500 mg/d in non-smokers and low alcohol consumers. We demonstrate that inverse associations are strongest for current smokers and individuals with high alcohol consumption, and that a lower risk is seen for higher intakes in these groups.

The present study has many strengths including a large adult population followed for 23 years with a plethora of participant characteristics captured at baseline. Another strength is the limited loss to follow-up; all deaths and emigrations are captured in the Danish registries. Furthermore, the data allowed us to adjust for many potential dietary and lifestyle confounders and our findings are strengthened with a falsification endpoint analysis, which showed no association with total flavonoid intake. Despite these strengths some limitations apply. As this is an observational study, we are not able to infer causality. The possibility of flavonoids being a marker of other unobserved and potentially protective dietary factors cannot be discounted, although, in this study the associations between total flavonoid intake and mortality rates remained even after adjusting for other major indicators of a healthy diet as well as within the highest tertile of total fruit and vegetable intake. As dietary intake data were only captured at baseline, this may have changed over the 23 years of follow-up. However, this misclassification would most likely have attenuated the power to detect an association. Common FFQ limitations apply in that not all flavonoid-rich foods, for example berries, were captured. Furthermore, potential confounders were only assessed at baseline; it is unclear how changes in the trajectories of these confounders may have impacted upon the observed associations. CVD- and cancer-related mortality

findings warrant further investigation as the Danish Register of Causes of Death are based on clinical adjudication and are susceptible to misclassification bias. For this reason, we only used broad definitions of cause-specific mortality[10]. Further caution should be taken when extrapolating these findings to populations outside of this cohort as the Danish population is more homogenous than many other countries; participants would most likely have been Caucasian.

While previous studies have demonstrated that higher flavonoid intake is associated with a lower risk of all-cause mortality, CVD mortality, and cancer mortality[11,12], this is the first observational study to show that this association is present for all subclasses after adjustment for lifestyle and dietary confounders. This is likely attributable to our large cohort size, high number of events, comprehensive estimation of flavonoid intake and high prevalence of smoking and high alcohol consumption.

The theory, that there exists a threshold after which higher intakes afford no added benefit is not new[12,13]. However, a critical gap in our knowledge has been what intake of total and specific flavonoids is required to achieve maximum benefit[14]. Results from our study indicate that for total flavonoid intake, risk of all-cause and CVD mortality was lower for flavonoid consumption until intakes of approximately 500 mg/d, after which higher intakes afforded no added benefit. This threshold was higher, approximately 1000 mg/d for cancer-related mortality. That the thresholds for each of the flavonoid subclasses approximately sum to the threshold for total flavonoid intake is consistent with the idea that all are important and afford added benefit. Interestingly, these threshold levels exist well within daily dietary achievable limits: one cup of tea, one apple, one orange, 100 g of blueberries, and 100 g of broccoli would provide most of the flavonoid subclasses and over 500 mg of total flavonoids. In this population it is likely that tea, chocolate, wine, apples, and pears were the main food sources of flavonoids[15].

We demonstrated that associations between intakes of flavonoids and both all-cause and cause-specific mortality were stronger in current smokers and in those who consume, on average, more than 20 g (two standard drinks) of alcohol per day. In the Iowa Women's Health Study there was evidence of an inverse association between intakes the flavonol and flavanone subclasses and all-cause mortality in ever-smokers, but not in never-smokers[16]. Cigarette smoking[17] and high alcohol consumption (>2 standard drinks per day)[18] are known to be carcinogenic and detrimental to endothelial function, blood pressure, inflammation, nitric oxide bioavailability, blood lipids, platelet function, and thrombosis. There is growing evidence from preclinical and intervention studies that flavonoids have growth inhibitory effects on various cancer cell types[19] and that they may ameliorate the above mentioned intermediate risk factors for early mortality[14,20,21]. This may explain the stronger association observed in smokers and heavy drinkers. Interestingly, and in accordance with our results, a meta-analysis of flavonoid intake showed an inverse association with smoking-related cancer incidence[22]. The implications that dietary flavonoid intake may partially mitigate harmful carcinogenic, metabolic, inflammatory, and vascular effects of smoking and excess alcohol consumption should not dissuade population health centers from anti-smoking or alcohol awareness campaigns. Even with high flavonoid consumption, these individuals remain at a much higher risk of early mortality. Rather it should be used to inform future clinical trials in these high-risk individuals and contribute to an optimization of dietary guidelines in these high-risk populations, ensuring that flavonoid intake is adequate and taking into consideration that these populations may benefit from higher intakes. Conversely, the inverse association between flavonoid intake and mortality tended to be weaker in obese participants. These findings go

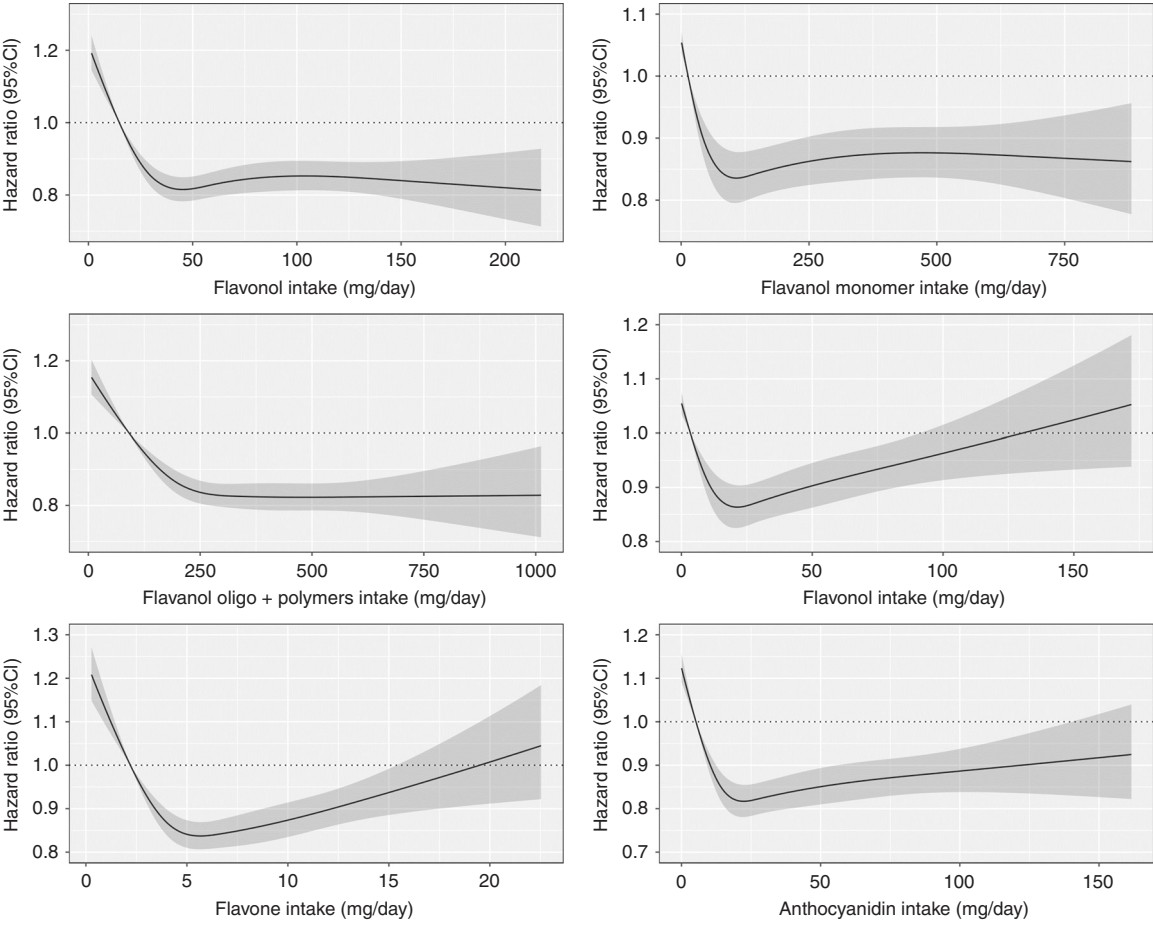

**Fig. 2** The association between flavonoid subclass intakes and all-cause mortality. Hazard ratios are based on Cox proportional hazards models adjusted for age, sex, BMI, smoking status, physical activity, alcohol intake, hypertension, hypercholesterolemia, social economic status (income), and prevalent disease and are comparing the specific level of flavonoid intake (horizontal axis) to the median intake for participants in the lowest intake quintile

against our a priori hypothesis that flavonoids would be more protective in obese individuals as they have higher levels of inflammation, oxidative stress, and vascular dysfunction. This is difficult to interpret, as there is evidence that flavonoids impact body composition[23] but may be explained in part by gut microbiome composition. The gut microbiome plays a critical role in flavonoid metabolism and therefore bioactivity[24], and is dissimilar in obese individuals[25]. Due to differences in baseline hazards between obese and non-obese participants, these findings warrant investigation on an absolute scale. In this Danish prospective cohort study, we demonstrate that a moderate intake of flavonoids is inversely associated with all-cause, CVD-related, and cancer-related mortality, with no added benefit seen for higher intakes in non-smokers and low alcohol consumers. The strongest associations observed between flavonoids and mortality was in smokers and high alcohol consumers, with higher intakes being more beneficial. These findings highlight the potential to improve population health through dietary recommendations to ensure adequate consumption of flavonoid-rich foods, particularly in these high-risk populations.

## Methods

**Study population.** From December 1993 to May 1997 the study recruited 56,468 participants without cancer prior to enrollment, between the age of 50–65 years, residing in the greater area of Copenhagen and Aarhus, Denmark. Details of the Danish Diet, Cancer, and Health cohort study, an associated cohort of the European Prospective Investigation into Nutrition and Cancer (EPIC), have been published previously[26]. All Danish residents are provided with a unique and

permanent civil registration number enabling crosslinking between nationwide registers and the Danish Diet, Cancer, and Health cohort on the individual level. The following databases were crosslinked to the cohort: The Civil Registration System which includes data on age, sex, and vital status; The Integrated Database for Labor Market Research which contains information on annual income since 1980; The Danish Register of Causes of death with information on cause of death since 1994 by International Classification of Diseases (ICD) codes; and The Danish National Patient Register (DNPR) which holds information on all hospital admissions in Denmark since 1978. All diagnoses were defined by ICD using the 8th revision (ICD-8) until 1993 and the 10th revision (ICD-10) since 1994.

Participants for whom information was missing or implausible ($n = 215$) and those with extreme energy intakes [<2 092 kJ/day (<500 kcal/day) and >20 920 kJ/day (>5000 kcal/day)] ($n = 205$), were excluded from the analysis (Fig. 4).

In Denmark, register studies do not require approval from ethics committees. The Danish Data Protection Agency approved this study (Ref no 2012–58–0004 I-Suite nr: 6357, VD-2018-117). Informed consent was obtained from all participants to search for information from medical registries.

**Dietary assessment.** Dietary data were collected using a validated 192-item food-frequency questionnaire, mailed out to participants prior to their baseline visit to one of the two study centers[27]. Participants were asked to indicate their usual frequency of intake of different food and beverage items over the past 12 months, using a 12-category frequency scale that ranged from never to 8 times or more per day. Details on the calculation of specific foods and nutrients have been published previously[26].

**Exposures.** The exposures of interest for this study were intakes of total flavonoids and flavonoid subclasses. An estimate of the flavonoid content of each food and beverage in the food-frequency questionnaire was derived from the Phenol-Explorer database[28]. Correlations between levels of 12 flavonoids in 24-hr urine samples and intake of their main food sources has been examined in 475 EPIC subjects, showing significant correlations with their main food sources[29]. The effect

**Table 2 Hazard ratios of all-cause mortality by quintiles of flavonoid intake**

| | Q1 (n = 11 210) | Q2 (n = 11 210) | Q3 (n = 11 209) | Q4 (n = 11 210) | Q5 (n = 11 209) |
|---|---|---|---|---|---|
| *Flavonols* | | | | | |
| No. events (%) | 3 778 | 2 964 | 2 615 | 2 418 | 2 245 |
| Intake (mg/d)[a] | 14.8 (0.1–20.5) | 25.9 (20.5–31.7) | 38.5 (31.7–49.7) | 66.0 (49.7–82.5) | 115.9 (82.5–250.7) |
| HR (95% CI) | | | | | |
| Model 1 | ref. | 0.79 (0.77, 0.82) | 0.68 (0.66, 0.71) | 0.64 (0.62, 0.67) | 0.64 (0.61, 0.67) |
| Model 2 | ref. | 0.86 (0.84, 0.89) | 0.80 (0.76, 0.83) | 0.81 (0.78, 0.85) | 0.83 (0.79, 0.87) |
| Model 3 | ref. | 0.89 (0.86, 0.92) | 0.84 (0.80, 0.87) | 0.87 (0.83, 0.91) | 0.90 (0.86, 0.95) |
| *Flavanol monomers* | | | | | |
| No. events (%) | 3 748 | 2 887 | 2 764 | 2 420 | 2 264 |
| Intake (mg/d)[a] | 13.5 (0–20.5) | 30.0 (20.5–45.3) | 66.2 (45.3–115.2) | 260.5 (115.2–281.4) | 473.3 (281.4–916.3) |
| HR (95% CI) | | | | | |
| Model 1 | ref. | 0.88 (0.86, 0.90) | 0.73 (0.70, 0.76) | 0.65 (0.62, 0.68) | 0.66 (0.63, 0.69) |
| Model 2 | ref. | 0.93 (0.91, 0.94) | 0.83 (0.80, 0.87) | 0.85 (0.81, 0.89) | 0.87 (0.83, 0.91) |
| Model 3 | ref. | 0.95 (0.93, 0.97) | 0.88 (0.84, 0.93) | 0.91 (0.87, 0.95) | 0.93 (0.89, 0.97) |
| *Flavanol oligo + polymers* | | | | | |
| No. events (%) | 3 680 | 2 969 | 2 633 | 2 510 | 2 291 |
| Intake (mg/d)[a] | 91 (0–136) | 179 (136–217) | 255 (217–302) | 365 (302–434) | 536 (360–2254) |
| HR (95% CI) | | | | | |
| Model 1 | ref. | 0.75 (0.73, 0.78) | 0.67 (0.64, 0.69) | 0.65 (0.62, 0.68) | 0.63 (0.60, 0.66) |
| Model 2 | ref. | 0.87 (0.84, 0.90) | 0.82 (0.79, 0.85) | 0.81 (0.78, 0.85) | 0.81 (0.77, 0.84) |
| Model 3 | ref. | 0.90 (0.87, 0.93) | 0.87 (0.83, 0.90) | 0.87 (0.83, 0.91) | 0.90 (0.86, 0.95) |
| *Flavanones* | | | | | |
| No. events (%) | 3 468 | 2 772 | 2 550 | 2 521 | 2 772 |
| Intake (mg/d)[a] | 3.4 (0–6.0) | 9.3 (6.0–12.7) | 17.5 (12.7–25.7) | 32.0 (25.8–49.0) | 70.1 (49.0–563.5) |
| HR (95% CI) | | | | | |
| Model 1 | ref. | 0.83 (0.80, 0.85) | 0.72 (0.69, 0.76) | 0.73 (0.70, 0.76) | 0.79 (0.76, 0.83) |
| Model 2 | ref. | 0.91 (0.88, 0.93) | 0.85 (0.81, 0.89) | 0.86 (0.82, 0.90) | 0.91 (0.87, 0.96) |
| Model 3 | ref. | 0.92 (0.89, 0.95) | 0.87 (0.83, 0.91) | 0.88 (0.85, 0.92) | 0.95 (0.9, 0.99) |
| *Flavones* | | | | | |
| No. events (%) | 3 457 | 2 835 | 2 560 | 2 456 | 2 775 |
| Intake (mg/d)[a] | 2.2 (0–3.0) | 3.7 (3.0–4.5) | 5.3 (4.5–6.1) | 7.2 (6.1–8.8) | 11.4 (8.8–50.6) |
| HR (95% CI) | | | | | |
| Model 1 | ref. | 0.76 (0.74, 0.78) | 0.67 (0.65, 0.70) | 0.68 (0.65, 0.70) | 0.73 (0.70, 0.76) |
| Model 2 | ref. | 0.88 (0.86, 0.91) | 0.84 (0.81, 0.87) | 0.85 (0.81, 0.88) | 0.89 (0.85, 0.94) |
| Model 3 | ref. | 0.90 (0.87, 0.92) | 0.86 (0.83, 0.90) | 0.88 (0.84, 0.92) | 0.95 (0.90, 1.00) |
| *Anthocyanins* | | | | | |
| No. events (%) | 3 599 | 2 521 | 2 426 | 2 693 | 2 844 |
| Intake (mg/d)[a] | 5.2 (0–9.6) | 12.5 (9.6–16.9) | 20.0 (16.9–24.4) | 35.7 (24.4–53.2) | 70.7 (53.2–396.9) |
| HR (95% CI) | | | | | |
| Model 1 | ref. | 0.71 (0.69, 0.74) | 0.63 (0.60, 0.65) | 0.69 (0.66, 0.72) | 0.80 (0.77, 0.84) |
| Model 2 | ref. | 0.84 (0.81, 0.87) | 0.78 (0.75, 0.82) | 0.80 (0.77, 0.84) | 0.83 (0.79, 0.87) |
| Model 3 | ref. | 0.85 (0.82, 0.88) | 0.80 (0.77, 0.84) | 0.82 (0.79, 0.86) | 0.86 (0.82, 0.90) |
| *Total flavonoids* | | | | | |
| No. events (%) | 3 706 | 2 900 | 2 728 | 2 455 | 2 294 |
| Intake (mg/d)[a] | 173.0 (5.6–250.8) | 320.2 (250.8–394.0) | 494.3 (394.1–601.2) | 725.9 (601.2–908.4) | 1201.3 (908.5–3552.0) |
| HR (95% CI) | | | | | |
| Model 1 | ref. | 0.78 (0.76, 0.81) | 0.68 (0.65, 0.71) | 0.66 (0.63, 0.69) | 0.64 (0.61, 0.67) |
| Model 2 | ref. | 0.88 (0.85, 0.91) | 0.83 (0.80, 0.86) | 0.83 (0.80, 0.87) | 0.83 (0.80, 0.87) |
| Model 3 | ref. | 0.91 (0.88, 0.94) | 0.88 (0.84, 0.91) | 0.90 (0.86, 0.94) | 0.92 (0.88, 0.97) |

Hazard ratios (95% CI) for 23-year all-cause mortality obtained from restricted cubic splines based on Cox proportional hazards models. Model 1 adjusted for age and sex; Model 2 adjusted for age, sex, BMI, smoking status, physical activity, alcohol intake, hypertension, hypercholesterolemia, social economic status (income), diabetes, and prevalent disease; Model 3 adjusted for all covariates in Model 2 plus intakes of fish, red meat, processed meat, dietary fiber, polyunsaturated fatty acids, monounsaturated fatty acids, and saturated fatty acids
[a]Median; range in parentheses (all such values)

of food processing on flavonoid content was taken into consideration using retention factors[30]. In the present study intakes of all flavonoids that were both available in Phenol-Explorer and could be estimated from foods in the FFQ (n = 219) were used (Supplementary Table 5). These were grouped into 10 subclasses according to their chemical structure [flavonols, flavanol monomers, flavanol oligo + polymers, flavanones, flavones, anthocyanins, isoflavones, dihydrochalcones, dihydroflavonols and chalcones] by summing the intakes of all individual flavonoid compounds within that flavonoid subclass (Supplementary Table 5). As the average intakes of isoflavones, dihydrochalcones, dihydroflavonols and chalcones were very low in this cohort (<5 mg/day), they were not included in the individual subclass analyses. Total flavonoid intake was calculated by summing all 219 flavonoid

compounds. The content of flavonoids was expressed as aglycones in mg/100 g fresh food weight.

**Study outcomes.** Vital status and date of death for every participant was obtained from the Civil Registration System. Cause of death data was obtained from the National Death Register. CVD-related mortality was defined as any ICD-10 diagnoses registered as a cause of death related to CVD (I00-I99) and cancer-related mortality was defined as any ICD-10 diagnosis registered as a cause of death related to cancer (C00-C99), dated after participant enrollment.

**Table 3 Hazard ratios of cardiovascular disease-related mortality by quintiles of flavonoid intake**

|  | Q1 | Q2 | Q3 | Q4 | Q5 |
|---|---|---|---|---|---|
| *Flavonols* | | | | | |
| No. events (%) | 1 112 | 844 | 712 | 689 | 624 |
| Intake (mg/d)[a] | 14.8 (0.1–20.5) | 25.9 (20.5–31.7) | 38.5 (31.7–49.7) | 66.0 (49.7–82.5) | 115.9 (82.5–250.7) |
| HR (95% CI) | | | | | |
| Model 1 | ref. | 0.76 (0.73, 0.80) | 0.64 (0.60, 0.69) | 0.62 (0.57, 0.67) | 0.62 (0.56, 0.67) |
| Model 2 | ref. | 0.84 (0.80, 0.88) | 0.77 (0.71, 0.83) | 0.82 (0.76, 0.89) | 0.85 (0.78, 0.93) |
| Model 3 | ref. | 0.88 (0.83, 0.93) | 0.84 (0.77, 0.91) | 0.91 (0.83, 0.99) | 0.95 (0.87, 1.05) |
| *Flavanol monomers* | | | | | |
| No. events (%) | 1 082 | 821 | 791 | 669 | 618 |
| Intake (mg/d)[a] | 13.5 (0–20.5) | 30.0 (20.5–45.3) | 66.2 (45.3–115.2) | 260.5 (115.2–281.4) | 473.3 (281.4–916.3) |
| HR (95% CI) | | | | | |
| Model 1 | ref. | 0.87 (0.84, 0.91) | 0.72 (0.66, 0.78) | 0.64 (0.59, 0.70) | 0.64 (0.59, 0.70) |
| Model 2 | ref. | 0.92 (0.89, 0.96) | 0.82 (0.76, 0.90) | 0.88 (0.81, 0.96) | 0.90 (0.82, 0.98) |
| Model 3 | ref. | 0.95 (0.92, 0.99) | 0.90 (0.82, 0.98) | 0.95 (0.87, 1.04) | 0.97 (0.89, 1.06) |
| *Flavanol oligo + polymers* | | | | | |
| No. events (%) | 1 050 | 849 | 743 | 718 | 621 |
| Intake (mg/d)[a] | 91 (0–136) | 179 (136–217) | 255 (217–302) | 365 (302–434) | 536 (360–2254) |
| HR (95% CI) | | | | | |
| Model 1 | ref. | 0.74 (0.69, 0.78) | 0.65 (0.61, 0.70) | 0.64 (0.59, 0.69) | 0.61 (0.56, 0.66) |
| Model 2 | ref. | 0.87 (0.82, 0.92) | 0.83 (0.77, 0.89) | 0.82 (0.76, 0.89) | 0.82 (0.75, 0.89) |
| Model 3 | ref. | 0.91 (0.85, 0.97) | 0.89 (0.82, 0.95) | 0.90 (0.83, 0.98) | 0.94 (0.85, 1.03) |
| *Flavanones* | | | | | |
| No. events (%) | 975 | 774 | 751 | 664 | 817 |
| Intake (mg/d)[a] | 3.4 (0–6.0) | 9.3 (6.0–12.7) | 17.5 (12.7–25.7) | 32.0 (25.8–49.0) | 70.1 (49.0–563.5) |
| HR (95% CI) | | | | | |
| Model 1 | ref. | 0.83 (0.78, 0.87) | 0.72 (0.66, 0.78) | 0.71 (0.66, 0.77) | 0.79 (0.73, 0.86) |
| Model 2 | ref. | 0.91 (0.86, 0.96) | 0.85 (0.77, 0.92) | 0.85 (0.78, 0.92) | 0.92 (0.84, 1.00) |
| Model 3 | ref. | 0.93 (0.88, 0.98) | 0.88 (0.81, 0.96) | 0.89 (0.82, 0.96) | 0.96 (0.88, 1.05) |
| *Flavones* | | | | | |
| No. events (%) | 956 | 823 | 676 | 720 | 806 |
| Intake (mg/d)[a] | 2.2 (0–3.0) | 3.7 (3.0–4.5) | 5.3 (4.5–6.1) | 7.2 (6.1–8.8) | 11.4 (8.8–50.6) |
| HR (95% CI) | | | | | |
| Model 1 | ref. | 0.76 (0.71, 0.80) | 0.67 (0.62, 0.72) | 0.68 (0.63, 0.73) | 0.74 (0.68, 0.81) |
| Model 2 | ref. | 0.89 (0.84, 0.95) | 0.85 (0.79, 0.92) | 0.86 (0.80, 0.93) | 0.92 (0.85, 1.00) |
| Model 3 | ref. | 0.91 (0.85, 0.96) | 0.87 (0.81, 0.94) | 0.89 (0.82, 0.97) | 0.97 (0.88, 1.06) |
| *Anthocyanins* | | | | | |
| No. events (%) | 1 051 | 694 | 683 | 769 | 784 |
| Intake (mg/d)[a] | 5.2 (0–9.6) | 12.5 (9.6–16.9) | 20.0 (16.9–24.4) | 35.7 (24.4–53.2) | 70.7 (53.2–396.9) |
| HR (95% CI) | | | | | |
| Model 1 | ref. | 0.67 (0.63, 0.71) | 0.58 (0.54, 0.63) | 0.66 (0.61, 0.72) | 0.77 (0.71, 0.84) |
| Model 2 | ref. | 0.81 (0.76, 0.86) | 0.75 (0.70, 0.82) | 0.81 (0.74, 0.87) | 0.83 (0.76, 0.91) |
| Model 3 | ref. | 0.85 (0.80, 0.91) | 0.81 (0.75, 0.88) | 0.86 (0.79, 0.94) | 0.90 (0.82, 0.99) |
| *Total flavonoids* | | | | | |
| No. events (%) | 1 062 | 818 | 773 | 710 | 618 |
| Intake (mg/d)[a] | 173.0 (5.6–250.8) | 320.2 (250.8–394.0) | 494.3 (394.1–601.2) | 725.9 (601.2–908.4) | 1201.3 (908.5–3552.0) |
| HR (95% CI) | | | | | |
| Model 1 | ref. | 0.77 (0.73, 0.82) | 0.67 (0.62, 0.72) | 0.66 (0.61, 0.71) | 0.62 (0.57, 0.67) |
| Model 2 | ref. | 0.88 (0.83, 0.93) | 0.84 (0.78, 0.90) | 0.87 (0.80, 0.94) | 0.85 (0.78, 0.93) |
| Model 3 | ref. | 0.92 (0.87, 0.98) | 0.91 (0.84, 0.98) | 0.96 (0.88, 1.04) | 0.97 (0.88, 1.06) |

Hazard ratios (95% CI) for 23-year cardiovascular disease-related mortality, in participants without cardiovascular disease at baseline (*n* = 52 492), obtained from restricted cubic splines based on Cox proportional hazards models. Model 1 adjusted for age and sex; Model 2 adjusted for age, sex, BMI, smoking status, physical activity, alcohol intake, hypertension, hypercholesterolemia, social economic status (income), diabetes, and prevalent disease; Model 3 adjusted for all covariates in Model 2 plus intakes of fish, red meat, processed meat, dietary fiber, polyunsaturated fatty acids, monounsaturated fatty acids, and saturated fatty acids
[a]Median; range in parentheses (all such values)

**Covariates**. Information on sex, age, and lifestyle factors such as smoking, alcohol consumption, and daily activity were obtained using the self-administered questionnaire at study enrollment. Clinical measurements such as BMI and total cholesterol were taken at the study centers. Annual income was used as a proxy for socio-economic status and was defined as household income after taxation and interest, for the value of the Danish currency in 2015. Income, grouped in quartiles, was estimated as the mean income of 5 years up to and including the year of study inclusion. Self-reported myocardial infarction and self-reported stroke at baseline were combined with ICD codes (Supplementary Table 6) of ischemic heart disease and ischemic stroke, respectively, dated prior to participant enrollment. ICD codes dated prior to participant enrollment were used to identify baseline comorbidities of peripheral artery disease, chronic kidney disease (CKD), chronic obstructive pulmonary disease (COPD), heart failure, atrial fibrillation, and cancers

(Supplementary Table 6). For hypertension and diabetes mellitus, only self-reported prevalence was used due to the low validity of ICD codes in the DNPR[31]. Prevalent CVD was defined by the presence of at least one diagnosis of ischemic heart disease, peripheral artery disease, ischemic stroke, heart failure, or atrial fibrillation prior to recruitment.

**Statistical analysis**. Participants were followed for a maximum of 23 years, from the date of enrollment until the date of death, emigration, or end of follow-up (August, 2017), whichever came first. The exposure variables (total flavonoid and flavonoid subclass intakes) were categorized by quintiles of intake (20% of participants from the total study population in each). Correlations between flavonoid subclasses were examined using Pearsons' correlation coefficients.

**Table 4 Hazard ratios of cancer-related mortality by quintiles of flavonoid intake**

|  | Q1 | Q2 | Q3 | Q4 | Q5 |
|---|---|---|---|---|---|
| *Flavonols* | | | | | |
| No. events (%) | 1 641 | 1 312 | 1 216 | 1 137 | 993 |
| Intake (mg/d)[a] | 14.8 (0.1–20.5) | 25.9 (20.5–31.7) | 38.5 (31.7–49.7) | 66.0 (49.7–82.5) | 115.9 (82.5–250.7) |
| HR (95% CI) | | | | | |
| Model 1 | ref. | 0.79 (0.75, 0.83) | 0.71 (0.67, 0.75) | 0.67 (0.63, 0.72) | 0.64 (0.60, 0.69) |
| Model 2 | ref. | 0.89 (0.85, 0.93) | 0.83 (0.78, 0.88) | 0.82 (0.77, 0.88) | 0.81 (0.75, 0.87) |
| Model 3 | ref. | 0.92 (0.88, 0.97) | 0.88 (0.83, 0.95) | 0.89 (0.83, 0.95) | 0.88 (0.81, 0.95) |
| *Flavanol monomers* | | | | | |
| No. events (%) | 1 650 | 1 338 | 1 198 | 1 093 | 1 020 |
| Intake (mg/d)[a] | 13.5 (0–20.5) | 30.0 (20.5–45.3) | 66.2 (45.3–115.2) | 260.5 (115.2–281.4) | 473.3 (281.4–916.3) |
| HR (95% CI) | | | | | |
| Model 1 | ref. | 0.88 (0.86, 0.91) | 0.73 (0.69, 0.78) | 0.66 (0.62, 0.70) | 0.65 (0.61, 0.69) |
| Model 2 | ref. | 0.92 (0.89, 0.95) | 0.82 (0.77, 0.88) | 0.83 (0.78, 0.89) | 0.83 (0.77, 0.89) |
| Model 3 | ref. | 0.95 (0.92, 0.98) | 0.88 (0.82, 0.94) | 0.88 (0.82, 0.94) | 0.88 (0.82, 0.94) |
| *Flavanol oligo + polymers* | | | | | |
| No. events (%) | 1 594 | 1 346 | 1 209 | 1 119 | 1 031 |
| Intake (mg/d)[a] | 91 (0–136) | 179 (136–217) | 255 (217–302) | 365 (302–434) | 536 (360–2254) |
| HR (95% CI) | | | | | |
| Model 1 | ref. | 0.79 (0.75, 0.83) | 0.71 (0.67, 0.75) | 0.67 (0.63, 0.72) | 0.64 (0.60, 0.69) |
| Model 2 | ref. | 0.90 (0.85, 0.94) | 0.85 (0.80, 0.90) | 0.81 (0.76, 0.87) | 0.79 (0.74, 0.85) |
| Model 3 | ref. | 0.92 (0.88, 0.97) | 0.88 (0.83, 0.94) | 0.86 (0.80, 0.92) | 0.86 (0.80, 0.93) |
| *Flavanones* | | | | | |
| No. events (%) | 1 521 | 1 299 | 1 170 | 1 145 | 1 164 |
| Intake (mg/d)[a] | 3.4 (0–6.0) | 9.3 (6.0–12.7) | 17.5 (12.7–25.7) | 32.0 (25.8–49.0) | 70.1 (49.0–563.5) |
| HR (95% CI) | | | | | |
| Model 1 | ref. | 0.86 (0.82, 0.89) | 0.76 (0.71, 0.81) | 0.73 (0.69, 0.78) | 0.75 (0.70, 0.80) |
| Model 2 | ref. | 0.91 (0.78, 0.95) | 0.84 (0.78, 0.90) | 0.82 (0.77, 0.87) | 0.83 (0.77, 0.88) |
| Model 3 | ref. | 0.92 (0.88, 0.97) | 0.87 (0.81, 0.93) | 0.84 (0.79, 0.90) | 0.85 (0.79, 0.91) |
| *Flavones* | | | | | |
| No. events (%) | 1 532 | 1 293 | 1 184 | 1 093 | 1 197 |
| Intake (mg/d)[a] | 2.2 (0–3.0) | 3.7 (3.0–4.5) | 5.3 (4.5–6.1) | 7.2 (6.1–8.8) | 11.4 (8.8–50.6) |
| HR (95% CI) | | | | | |
| Model 1 | ref. | 0.79 (0.75, 0.82) | 0.69 (0.66, 0.73) | 0.68 (0.64, 0.72) | 0.71 (0.67, 0.76) |
| Model 2 | ref. | 0.88 (0.84, 0.92) | 0.82 (0.78, 0.87) | 0.81 (0.76, 0.86) | 0.84 (0.78, 0.90) |
| Model 3 | ref. | 0.88 (0.84, 0.92) | 0.83 (0.78, 0.88) | 0.82 (0.77, 0.87) | 0.85 (0.79, 0.92) |
| *Anthocyanins* | | | | | |
| No. events (%) | 1 538 | 1 176 | 1 124 | 1 215 | 1 246 |
| Intake (mg/d)[a] | 5.2 (0–9.6) | 12.5 (9.6–16.9) | 20.0 (16.9–24.4) | 35.7 (24.4–53.2) | 70.7 (53.2–396.9) |
| HR (95% CI) | | | | | |
| Model 1 | ref. | 0.76 (0.73, 0.80) | 0.69 (0.65, 0.74) | 0.76 (0.71, 0.81) | 0.83 (0.78, 0.89) |
| Model 2 | ref. | 0.87 (0.83, 0.91) | 0.82 (0.77, 0.87) | 0.83 (0.78, 0.89) | 0.83 (0.78, 0.90) |
| Model 3 | ref. | 0.91 (0.86, 0.96) | 0.88 (0.82, 0.94) | 0.89 (0.83, 0.96) | 0.90 (0.83, 0.97) |
| *Total flavonoids* | | | | | |
| No. events (%) | 1 607 | 1 348 | 1 240 | 1 083 | 1 021 |
| Intake (mg/d)[a] | 173.0 (5.6–250.8) | 320.2 (250.8–394.0) | 494.3 (394.1–601.2) | 725.9 (601.2–908.4) | 1201.3 (908.5–3552.0) |
| HR (95% CI) | | | | | |
| Model 1 | ref. | 0.81 (0.78, 0.85) | 0.71 (0.67, 0.75) | 0.67 (0.63, 0.71) | 0.64 (0.59, 0.68) |
| Model 2 | ref. | 0.90 (0.86, 0.94) | 0.84 (0.79, 0.89) | 0.81 (0.76, 0.87) | 0.80 (0.74, 0.85) |
| Model 3 | ref. | 0.93 (0.88, 0.97) | 0.88 (0.83, 0.94) | 0.86 (0.80, 0.92) | 0.86 (0.80, 0.93) |

Hazard ratios (95% CI) for 23-year cancer-related mortality, in participants without cancer at baseline ($n = 55\ 801$), obtained from restricted cubic splines based on Cox proportional hazards models. Model 1 adjusted for age and sex; Model 2 adjusted for age, sex, BMI, smoking status, physical activity, alcohol intake, hypertension, hypercholesterolemia, social economic status (income), diabetes, and prevalent disease; Model 3 adjusted for all covariates in Model 2 plus intakes of fish, red meat, processed meat, dietary fiber, polyunsaturated fatty acids, monounsaturated fatty acids, and saturated fatty acids
[a]Median; range in parentheses (all such values)

Potential nonlinear relationships were examined using restricted cubic splines, with hazard ratios (HRs) based on Cox proportional hazards models. HRs with 95% confidence intervals (CIs) were plotted for each unit of the exposure against the median intake in quintile 1. In all spline analyses, the exposure variables were treated as continuous and individuals with intakes more than four standard deviations above the mean were excluded. The test of nonlinearity used analysis of variance to compare the model with only the linear term to the model that included both the linear and the cubic spline terms. HRs and 95% CIs for the median intakes in each quintile of the exposure variables were obtained from the splines. Cox proportional hazards assumptions were tested using log-log plots of the survival function vs. time and assessed for parallel appearance. As our aim was to obtain relative estimates for risk factors, deaths from causes other than the outcome of interest were censored rather than

treated as a competing risk[32]. Three models of adjustment were used: (1) age and sex; (2) age, sex, BMI, smoking status, physical activity (total daily metabolic equivalent), pure alcohol intake (g/d), hypertension (yes/no), hypercholesterolemia (yes/no), social economic status (income), and prevalent disease (diabetes, CVD, COPD, CKD and cancer, entered into the model separately); (3) model 2 plus intakes (g/d) of fish, red meat, processed meat, dietary fiber, polyunsaturated fatty acids, monounsaturated fatty acids and saturated fatty acids. Covariates were chosen a priori to the best of our knowledge of potential confounders of flavonoid intake and mortality. As some covariates in model 2 (hypertension, hypercholesterolemia and prevalent diseases) are potentially on the causal pathway and therefore introduce collider stratification bias, we removed them in a sensitivity analysis. When investigating CVD-related mortality we excluded participants with CVD at baseline. Using

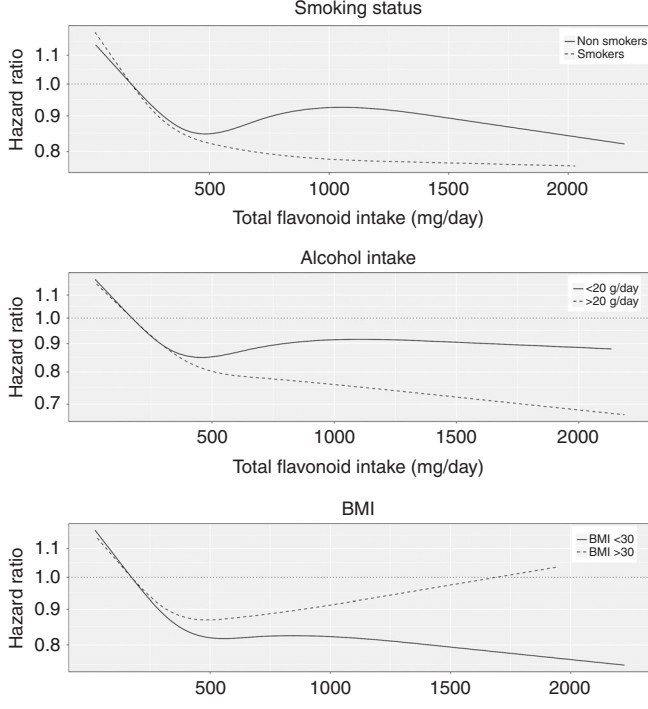

**Fig. 3** The association of total flavonoid intake with all-cause mortality, stratified by risk factors. Multivariable-adjusted association between total flavonoid intake and all-cause mortality stratified by current smoking status, alcohol intake and BMI. Values are hazards ratios and 95% CI for the highest compared to the lowest quintiles of intake. All analyses were standardized for age, sex, BMI, physical activity, alcohol intake, hypertension, hypercholesterolemia, smoking status, social economic status (income), and prevalent disease, not including the stratification variable for the subgroups

ICD-8 codes, we identified an additional 247 participants with a diagnosis of cancer prior to enrollment; when investigating cancer-related mortality we excluded these participants (Fig. 4).

We stratified our analyses by sex, BMI, smoking history, alcohol intake, physical activity, and prevalent diabetes to test for potential effect modification. When stratifying by alcohol intake and BMI, we excluded all participants with an alcohol intake of zero ($n = 1\,298$) and a BMI < 18.5 ($n = 453$) respectively, as these were not our subgroups of interest. We chose stratification cut-off points of 20 g pure alcohol per day, corresponding to 2 standard drinks, and a BMI of 30 kg/m$^2$ as the risk of mortality is highest beyond these levels[18,33]. A further subgroup 'dose–response' analysis was performed by pack-years of smoking, alcohol intake and BMI categories.

To account for the possibility of differing dietary habits in those at a high risk of death, we repeated our primary analysis after excluding participants with prevalent diabetes, CVD, COPD, CKD, and cancer. As we believe crude values of flavonoid intake to be more relevant than energy-adjusted values, we did not include total energy intake as a covariate in any model. However, energy intake was added to model 2 in a sensitivity analysis to assess its impact on the association between flavonoid intake and all-cause mortality. To determine whether the associations between flavonoid intake and all-cause mortality were independent of fruit and vegetable consumption, we stratified the analysis by tertiles of total fruit and vegetable intake. In order to assess the likelihood of confounding we used a falsification endpoint which we considered unlikely to be causally affected by flavonoid intake; any emergency, inpatient, or outpatient visit for a burn or foreign object (Supplementary Table 6). Analyses were undertaken using STATA/IC 14.2 (StataCorp LLC) and R statistics (R Core Team[34]). Statistical significance was set at $p \le 0.05$ (two-tailed) for all tests.

**Role of funding source.** The funding source had no role in study design, preparation of this manuscript, or decision to submit the paper for publication.

**Reporting summary.** Further information on research design is available in the Nature Research Reporting Summary linked to this article.

## Data availability

Due to restrictions related to Danish law and protecting patient privacy, the combined set of data as used in this study can only be made available through a trusted third party, Statistics Denmark. This state organization holds the data used for this study. University-based Danish scientific organizations can be authorized to work with data within Statistics Denmark and such organization can provide access to individual scientists inside and outside of Denmark. Requests for data may be sent to Statistics Denmark: https://www.dst.dk/en/kontakt or the Danish Data Protection Agency: https://www.datatilsynet.dk/english.

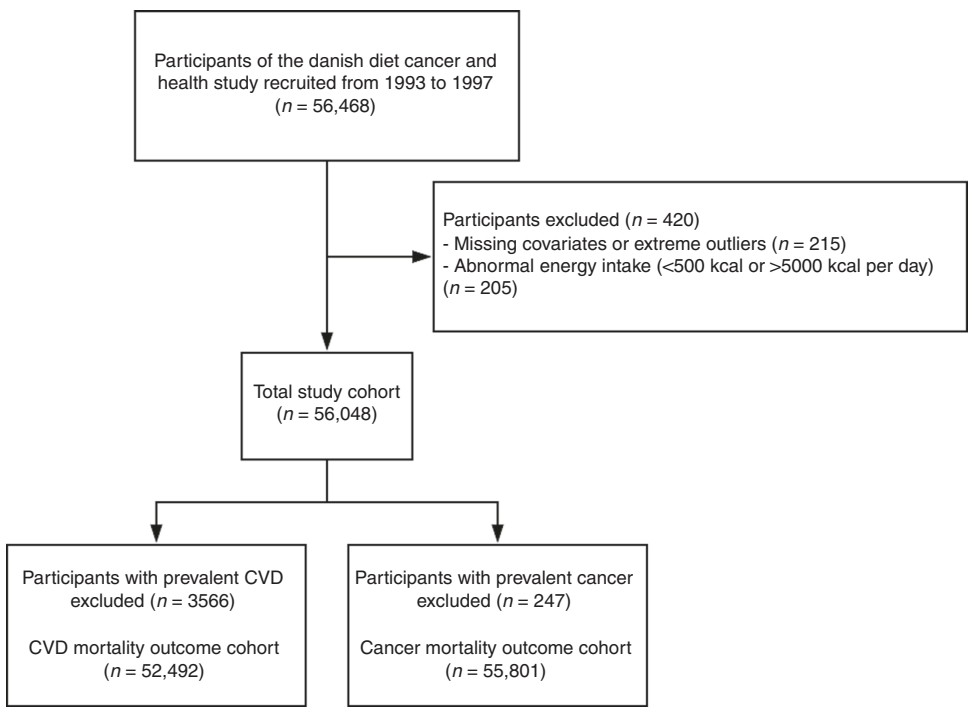

**Fig. 4** Consort flow diagram. *CVD* cardiovascular disease

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

## Acknowledgements

The authors wish to thank Professor Thomas A. Gerds for his statistical advice. The Danish Diet, Cancer, and Health Study was funded by the Danish Cancer Society, Denmark. FD is funded by The Danish Heart Foundation (Grant number 17-R115-A7443-22062) and Gangstedfonden (Grant number A35136), Denmark. NPB is funded by a National Health and Medical Research Council Early Career Fellowship (Grant number APP1159914), Australia. The salary of JMH is supported by a National Health and Medical Research Council of Australia Senior Research Fellowship, Australia (Grant number APP1116937).

## Disclaimer

Where authors are identified as personnel of the International Agency for Research on Cancer/World Health Organization, the authors alone are responsible for the views expressed in this article and they do not necessarily represent the decisions, policy or views of the International Agency for Research on Cancer / World Health Organization.

## Author contributions

N.P.B., F.D., K.O., A.T., and J.M.H. contributed to the study concept and design; AS calculated the flavonoid intake from FFQ data; N.P.B. and F.D. conducted the data analysis; N.P.B. and F.D. drafted the manuscript; N.P.B., F.D., C.K., K.M., C.P.B., J.R.L., K.D.C., G.G., A.S., A.C., A.T., K.O., and J.M.H. critically reviewed the final draft of the manuscript.

## Additional information

**Competing interests:** The authors declare no competing interests.

**Peer Review Information:** *Nature Communications* thanks the anonymous reviewer(s) for their contribution to the peer review of this work. Peer reviewer reports are available.

