## [Peer Review File · Nature Communications]

Reviewers' comments:

Reviewer #1 (Remarks to the Author):

This is an analysis of flavonoid intake and mortality in the Danish Diet, Cancer and Health cohort study including 56000 participants and 14083 deaths. The study found an inverse association between intake of flavonoids and all-cause, CVD and cancer mortality, with nonlinear associations observed up to around 500 mg/d. The inverse association was more apparent in smokers and alcohol consumers. This is a well conducted and written article. I only have a few mostly minor comments below.

Suggest to add number of deaths in the abstract. Perhaps also say something about the strength of the observed associations?

line 198 - 208: say something about the main food sources of flavonoids in this study.

line 250-255: I would suggest to add the results from these sensitivity analyses to a supplementary table.

line 270-271: residual confounding by smoking is a possibility though since the associations were stronger in smokers.

line 297-298: it would have been interesting to know if any of these food sources were particularly beneficial.

Supplementary Table 2. Some other causes of death are listed than just CVD and cancer. Would it be possible to also analyse flavonoids and some other causes of death like respiratory disease, infections, neurological, kidney disease etc.? That could bring some more novelty to the paper. There are many studies already which have looked at CVD and cancer, but not many that have looked at other causes of death.

Reviewer #2 (Remarks to the Author):

Summary

This large study addresses the association between flavonoid intake and overall as well as cardiovascular and cancer specific mortality. The study was conducted in a large cohort in Denmark with 56,048 individuals aged 50-65 at baseline followed for 23 years. The large study size is a major strength of the study as is the availability of the population databases to determine outcomes as well as some comorbidities and income. Information on multiple potential confounders was obtained at baseline along with dietary intake from a food frequency questionnaire (FFQ). A novel aspect of the study was the assessment of the subclasses of flavonoids in a large cohort. In general, the methods used were fairly straightforward and reasonable. There are some questions about the methods and interpretation.

Comments

1. The cohort was defined as being without cancer at baseline (p.5), but subsequently in the paper there is mention of cases of prevalent cancer at baseline. This should be explained/clarified.
2. The FFQ is described as validated (p.6), but is there any information on the validity of the flavonoid assessment? Some information on the validation would be helpful.
3. The timing of assessment of diet and covariates should be clarified in the methods. It appears that most variables were assessed once at baseline. For comorbidities assessed by linkage to administrative databases, were these also based on status at baseline?
4. Is there any information on the validity of cause-specific mortality?
5. Hypertension and diabetes were accounted for based on self-report. Was the treatment for these conditions also accounted for? The authors should consider the potential influence of undiagnosed hypertension and diabetes in the interpretation of the results. In particular is the concern that undiagnosed disease may be more prevalent at the lower quintiles of flavonoid intake.
6. Was any treatment information (e.g. statins) considered in any models? These were included in the descriptive tables, but not mentioned in the methods or results. Was there a reason for not including them? How might not accounting for treatment of comorbidities affect the interpretation?
7. It should be noted that adding dietary variables to the model did attenuate the results somewhat.
8. On p. 12, although supplementary figure 5 does show a trend of greater reductions in overall and cancer mortality at higher levels of smoking or alcohol intake, this is not so clear for cardiovascular mortality

9. The discussion focuses on improving flavonoid intake, but how might this differ from current health promotion efforts to increase vegetable and fruit intake? It would be good to see how an analysis of vegetable and fruit intake compares to the analyses of flavonoids with respect to mortality. Is there a benefit to making recommendations specifically around flavonoids rather than vegetables and fruit which may provide multiple beneficial components (fibre and micronutrients in addition to flavonoids)?
10. One of the novel features of the study is the examination of the subclasses of flavonoids. However, the estimated effect sizes for the subclasses appeared to be all essentially the same. Is this what would be expected biologically? It would be good to see a bit more discussion of the subclass results.
11. It is not clear why residual confounding is unlikely.
12. The concern about having measurements at baseline only is not just that diet intake might change (and attenuate results), but also that potential confounders might change over time. For example, physical activity trajectories might differ in a manner associated with flavonoid intake and not be adequately captured by a baseline physical activity measure. It can be quite complex to tease apart multiple correlated factors and this complexity should at least be acknowledged.
13. The suggestion to develop high flavonoid content foods assumes causality, which may be premature. Also, caution is needed to ensure no harms are caused by radically increasing intake of any food component. Consider the controversies around folate fortification.
14. In table 1, since the proportion of women varies by quintile, the HRT distribution should be given for women only. Also in this table, it is not clear why aspirin is listed separately from NSAIDs since aspirin is an NSAID.
15. It is surprising that those in the lowest quintile of flavonoid intake have both a higher BMI (measured) and a lower energy intake (self-report from FFQ). Are there any concerns about differences in the accuracy of the FFQ across quintiles?
16. In figure 4 and supplementary figures 3 and 4, it is confusing and a bit misleading that for smoking and alcohol the healthier group is represented by a solid line and the less healthy group by a dotted line, but this situation is switched for BMI. I suggest that the BMI groups be switched with respect to group representation in the figures for consistency and ease of interpretation.
17. One thing that would make the paper and conclusions stronger would be the inclusion of a negative control analysis. This would be an analysis of total flavonoid intake with a cause of mortality not expected to be associated with flavonoids, such as accidents and injuries. If there is little suggestion of a relationship then it strengthens the conclusion of flavonoids contributing to reduced mortality from CVD and cancer. However, if similar estimated effects are observed it suggests that results may be driven at least in part by confounding or other bias.

The authors would like to thank the reviewers for taking the time to read through the manuscript and provide valuable feedback and comments. We have now revised the manuscript to address these comments.

Below is given a point-by-point response to the reviewer’s comments highlighted in bold and our answers are formatted in italic.

All indicated page numbers in this document refer to the manuscript file with highlighted (tracked) changes.

Reviewer 1

Qu	Reviewer comment/Author response	Page number
1	Suggest to add number of deaths in the abstract. Perhaps also say something about the strength of the observed associations?	
	Thank you for this suggestion. The abstract has now been modified to include the total number of deaths and the strength of the association. Other changes to the abstract were to ensure that we did not exceed the word limit stipulated by the journal.	3
2	line 198 - 208: say something about the main food sources of flavonoids in this study.	
	We agree with the reviewer that this is of interest. However, for this study, we used pre-calculated flavonoid intake data that were already available for the Danish Diet Cancer and Health cohort as it is part of the larger European Prospective Investigation into Nutrition and Cancer. For this reason, we were unable to directly ascertain which were the main food sources of flavonoids in this study. From a previous study published by Zamora-ros et al, we can see that apples and pears, chocolate, wine, and tea were the main sources of flavonoids. As this is not a result per se from the present study, we have included the following sentence in the discussion: “In this population it is likely that tea, chocolate, wine, apples, and pears were the main food sources of flavonoids¹⁵.” Ref 15: Zamora-Ros, R. et al. Dietary polyphenol intake in Europe: the European Prospective Investigation into Cancer and Nutrition (EPIC) study. European journal of nutrition 55, 1359-1375 (2016).	10
3	line 250-255: I would suggest to add the results from these sensitivity analyses to a supplementary table.	
	We have added the results from these sensitivity analyses to a new Supplementary Table 2 as shown below:	8

		Supplemental Table 2. Hazard ratios of all-cause mortality by quintiles of total flavonoid intake using alternative models of adjustment																																				
		Total flavonoid intake quintiles																																				
		Q1	Q2	Q3	Q4	Q5																																
		Model 2	ref.	0.88 (0.85, 0.91)	0.83 (0.80, 0.86)	0.83 (0.80, 0.87)	0.83 (0.80, 0.87)																															
		Model 2b	ref.	0.88 (0.85, 0.91)	0.83 (0.80, 0.86)	0.83 (0.80, 0.87)	0.83 (0.79, 0.87)																															
		Model 2c	ref.	0.88 (0.85, 0.90)	0.82 (0.79, 0.85)	0.82 (0.79, 0.86)	0.83 (0.79, 0.87)																															
		Model 2d	ref.	0.88 (0.86, 0.91)	0.83 (0.80, 0.86)	0.83 (0.80, 0.87)	0.83 (0.80, 0.87)																															
		Model 2*	ref.	0.88 (0.85, 0.91)	0.82 (0.79, 0.86)	0.82 (0.78, 0.86)	0.82 (0.78, 0.86)																															
		Hazard ratios (95% CI) for 23-year all-cause mortality obtained from restricted cubic splines based on Cox proportional hazards models. Model 2 (original model) adjusted for age, sex, BMI, smoking status, physical activity, alcohol intake, hypertension, hypercholesterolemia, social economic status (income), diabetes and prevalent disease; Model 2b: Model 2 plus energy intake; Model 2c: Model 2 without adjustment for prevalent diseases which may potentially lie on the causal pathway (adjusted for age, sex, BMI, smoking status, physical activity, alcohol intake, and, social economic status (income) only); Model 2d: Model 2 plus medication use (statins, antihypertensive medication, insulin and aspirin). *Analysis run excluding n=5 492 participants with comorbidities at baseline.																																				
4	line 270-271: residual confounding by smoking is a possibility though since the associations were stronger in smokers.																																					
	We agree with the reviewer that there may be residual confounding by smoking. We have investigated this further by stratifying the analysis by smoking intensity (pack-years). A higher smoking intensity is associated with a lower hazard ratio for participants in quintile 5, strongly suggesting that there is an interaction between flavonoid intake and smoking intensity in a dose-response manner. This analysis is included in	7																																				
	A. Smoking    Subgroups [Q5 vs Q1] Adjusted Hazard Ratio [95% CI]     All-cause mortality   Non-smokers 1.04 [0.92-1.17]   <15 packyears 0.87 [0.75-0.99]   15-30 packyears 0.79 [0.71-0.88]   >30 packyears 0.77 [0.69-0.84]   CVD mortality   Non-smokers 1.11 [0.88-1.40]   <15 packyears 0.80 [0.61-1.04]   15-30 packyears 0.71 [0.57-0.87]   >30 packyears 0.83 [0.70-0.98]   Cancer mortality   Non-smokers 1.09 [0.91-1.31]   <15 packyears 0.94 [0.76-1.15]   15-30 packyears 0.73 [0.62-0.86]   >30 packyears 0.72 [0.62-0.83]    Hazard Ratio [95% CI]	Subgroups [Q5 vs Q1]	Adjusted Hazard Ratio [95% CI]	All-cause mortality		Non-smokers	1.04 [0.92-1.17]	<15 packyears	0.87 [0.75-0.99]	15-30 packyears	0.79 [0.71-0.88]	>30 packyears	0.77 [0.69-0.84]	CVD mortality		Non-smokers	1.11 [0.88-1.40]	<15 packyears	0.80 [0.61-1.04]	15-30 packyears	0.71 [0.57-0.87]	>30 packyears	0.83 [0.70-0.98]	Cancer mortality		Non-smokers	1.09 [0.91-1.31]	<15 packyears	0.94 [0.76-1.15]	15-30 packyears	0.73 [0.62-0.86]	>30 packyears	0.72 [0.62-0.83]	Supplementary Figure 5.				
Subgroups [Q5 vs Q1]	Adjusted Hazard Ratio [95% CI]																																					
All-cause mortality																																						
Non-smokers	1.04 [0.92-1.17]																																					
<15 packyears	0.87 [0.75-0.99]																																					
15-30 packyears	0.79 [0.71-0.88]																																					
>30 packyears	0.77 [0.69-0.84]																																					
CVD mortality																																						
Non-smokers	1.11 [0.88-1.40]																																					
<15 packyears	0.80 [0.61-1.04]																																					
15-30 packyears	0.71 [0.57-0.87]																																					
>30 packyears	0.83 [0.70-0.98]																																					
Cancer mortality																																						
Non-smokers	1.09 [0.91-1.31]																																					
<15 packyears	0.94 [0.76-1.15]																																					
15-30 packyears	0.73 [0.62-0.86]																																					
>30 packyears	0.72 [0.62-0.83]																																					
5	line 297-298: it would have been interesting to know if any of these food sources were particularly beneficial.																																					
	Almost all dietary components are correlated, making it impossible to tease apart the benefit of one food in particular. We acknowledge that this limitation, common amongst nearly all nutritional epidemiology studies, is also true for flavonoids. However, looking at total flavonoid and flavonoid subclass intakes is more reflective of a dietary pattern rich in flavonoids. We have endeavoured to																																					

	play to the strengths of epidemiological research and thus the focus of the manuscript has been to demonstrate that the association between a flavonoid-rich diet and mortality is 1. non-linear and 2. stronger in smokers and those who consume high levels of alcohol. These findings are novel, hypothesis generating, and can be investigated in future randomised controlled trials.	
6	Supplementary Table 2. Some other causes of death are listed than just CVD and cancer. Would it be possible to also analyse flavonoids and some other causes of death like respiratory disease, infections, neurological, kidney disease etc.? That could bring some more novelty to the paper. There are many studies already which have looked at CVD and cancer, but not many that have looked at other causes of death.	
	We agree with the reviewer that the associations between flavonoid intake and other outcomes would be of interest. Supplementary Table 2 (now Supplementary Table 6) contains the ICD codes used for diagnosis of hospitalisation in the Danish National Patient Registry and cause of death in the Danish Register of Causes of Death. The cause-specific death in the death registry is adjudicated by the clinician at the hospital and rarely verified by autopsy. Therefore, there is a risk of misclassification bias. The clinical adjudication of ICD-codes should only be used as a broad indication to cause of death. This is a limitation of the registry. Another issue is that we defined our cause-specific analysis a priori based on our predefined hypothesis that flavonoids can mitigate CVD and cancer risks. Therefore, we have only chosen to look at the two causes of death hypothesized to be associated with flavonoid intake using a very broad definition for each outcome to reduce misclassification bias. As referenced in the methods, the registry of cause-specific death should be used with caution and only for broadly defined terms such as cancer and CVD mortality. We have added a sentence to the limitations that these findings warrant further investigation due to lacking validity and inability to specify the exact causes of death more than a broad definition of death: “CVD- and cancer-related mortality findings warrant further investigation as the Danish Register of Causes of Death are based on clinical adjudication and are prone to misclassification bias. For this reason, we only used broad definitions of cause-specific mortality¹⁰”. Ref 10: Helweg-Larsen, K. The Danish register of causes of death. Scand J Public Health 39, 26-29 (2011).	9 - 10

Reviewer 2

Q u	Reviewer comment/Author response	Page numbe r
1	The cohort was defined as being without cancer at baseline (p.5), but subsequently in the paper there is mention of cases of prevalent cancer at baseline. This should be explained/clarified.	
	We thank the reviewer for pointing out the confusion. In order not to invite participants with a prevalent cancer, a record linkage to the Danish Cancer Registry was made before an invitation to participate was sent. Due to a time lag to update the Cancer Registry or a diagnosed cancer between the invitation and the appointment at the study centre, some people with a very recent cancer were invited, and if they agreed to participate, subsequently included in the cohort. When searching for cancer-related ICD-8 codes in the Danish National Patient Register, we identified a further 247 cases of cancer prior to baseline and therefore excluded those participants as well. We have explained by adding the following to the manuscript: “Using ICD-8 codes, we identified an additional 247 participants with a diagnosis of cancer prior to enrolment; when investigating cancer-related mortality we excluded these participants (Figure 4).”	16
2	The FFQ is described as validated (p.6), but is there any information on the validity of the flavonoid assessment? Some information on the validation would be helpful.	
	Flavonoid assessment has not been systematically validated, due to the lack of suitable biomarkers of flavonoid intake for all flavonoids. However, correlations between levels of 34 polyphenols (including 12 flavonoids) in 24-hr urine samples and intake of their main food sources has been examined in 475 EPIC subjects, showing significant correlations with their main food sources²⁹. We also examined correlations between intake of some polyphenols and their levels in urine in the same subjects. Weak partial Spearman correlations between both dietary acute and habitual intake and urinary concentrations of quercetin, naringenin and hesperetin were observed (two manuscripts in preparation). The food composition table for flavonoids in the EPIC cohort is one of the most advanced tools to have been developed to measure polyphenol intake and one of a very few for which the effect of food processing on flavonoid content is taken into consideration using retention factors. The development of the food composition table has been described in detail in a recent publication³⁰ and these citations have been added to the manuscript as follows: “Correlations between levels of 12 flavonoids in 24-hr urine samples and intake of their main food sources has been examined in 475 EPIC subjects, showing significant correlations with their main food sources²⁹. The effect of food processing on flavonoid content was taken into consideration using retention factors³⁰.” Ref 29: Zamora-Ros, R. et al. Urinary excretions of 34 dietary polyphenols and their	14

	associations with lifestyle factors in the EPIC cohort study. Scientific reports 6, 26905 (2016). Ref 30: Knaze, V. et al. A new food-composition database for 437 polyphenols in 19,899 raw and prepared foods used to estimate polyphenol intakes in adults from 10 European countries. The American journal of clinical nutrition 108, 517-524 (2018).	
3	The timing of assessment of diet and covariates should be clarified in the methods. It appears that most variables were assessed once at baseline. For comorbidities assessed by linkage to administrative databases, were these also based on status at baseline?	
	Thank you for pointing out that the timing of these assessments was missing from the manuscript. We have clarified this by adding the underlined words as follows: “Dietary data were collected using a validated 192-item food-frequency questionnaire, mailed out to participants prior to their baseline visit to one of the two study centres”. “Self-reported myocardial infarction and self-reported stroke at baseline were combined with ICD codes (Supplementary Table 6) of ischemic heart disease and ischemic stroke, respectively, dated prior to participant enrolment. ICD codes dated prior to participant enrolment were used to identify baseline comorbidities of peripheral artery disease, chronic kidney disease (CKD), chronic obstructive pulmonary disease (COPD), heart failure, atrial fibrillation, and cancers (Supplementary Table 6)”.	13 & 15
4	Is there any information on the validity of cause-specific mortality?	
	We have added the following sentence to the limitations section of the discussion: “CVD- and cancer-related mortality findings warrant further investigation as the Danish Register of Causes of Death are based on clinical adjudication and are susceptible to misclassification bias. For this reason, we only used broad definitions of cause-specific mortality¹⁰”. Ref 10: Helweg-Larsen, K. The Danish register of causes of death. Scand J Public Health 39, 26-29 (2011).	9 - 10
5	Hypertension and diabetes were accounted for based on self-report. Was the treatment for these conditions also accounted for? The authors should consider the potential influence of undiagnosed hypertension and diabetes in the interpretation of the results. In particular is the concern that undiagnosed disease may be more prevalent at the lower quintiles of flavonoid intake.	
	We agree with the reviewer that there could be some undiagnosed disease in the cohort. Unfortunately, we are unable to capture undiagnosed diabetes as there were no measures of a glucose tolerance test or HbA1c. However, the inverse associations between flavonoid intake and mortality were already seen in quintile 2, where the participants were more similar to those in quintile 1 in terms of baseline characteristics (including the potential for undiagnosed hypertension and diabetes). In a sensitivity analysis we adjusted for antihypertensive and diabetic medication use. Please see our response to suggestion 6 below.	

6	Was any treatment information (e.g. statins) considered in any models? These were included in the descriptive tables, but not mentioned in the methods or results. Was there a reason for not including them? How might not accounting for treatment of comorbidities affect the interpretation?																																										
	We did not adjust for medication use in the main models as we did not hypothesize a priori that it was related to flavonoid intake other than through socio-economic status (i.e. educational status and affordability could influence the use of medication), which was already adjusted for. However, we have re-run the association between total flavonoid intake and all-cause mortality now adjusting for diabetes medication, antihypertensive medication, statins, and aspirin use in a sensitivity analysis and have included the results in a new Supplementary Table 2 (shown below). Adjusting for medication use did not change the association between total flavonoid intake and all-cause mortality (see Model 2d). We have included the following in the manuscript: “Neither including energy intake in model 2, nor removing hypertension, hypercholesterolemia and prevalent diseases from model 2, or adjusting for medication use substantively altered the hazard ratios (Supplementary Table 2)”. Supplemental Table 2. Hazard ratios of all-cause mortality by quintiles of total flavonoid intake using alternative models of adjustment     Total flavonoid intake quintiles   Q1 Q2 Q3 Q4 Q5     Model 2 ref. 0.88 (0.85, 0.91) 0.83 (0.80, 0.86) 0.83 (0.80, 0.87) 0.83 (0.80, 0.87)   Model 2b ref. 0.88 (0.85, 0.91) 0.83 (0.80, 0.86) 0.83 (0.80, 0.87) 0.83 (0.79, 0.87)   Model 2c ref. 0.88 (0.85, 0.90) 0.82 (0.79, 0.85) 0.82 (0.79, 0.86) 0.83 (0.79, 0.87)   Model 2d ref. 0.88 (0.86, 0.91) 0.83 (0.80, 0.86) 0.83 (0.80, 0.87) 0.83 (0.80, 0.87)   Model 2* ref. 0.88 (0.85, 0.91) 0.82 (0.79, 0.86) 0.82 (0.78, 0.86) 0.82 (0.78, 0.86)    <small>Hazard ratios (95% CI) for 23-year all-cause mortality obtained from restricted cubic splines based on Cox proportional hazards models. Model 2 (original model) adjusted for age, sex, BMI, smoking status, physical activity, alcohol intake, hypertension, hypercholesterolemia, social economic status (income), diabetes and prevalent disease; Model 2b: Model 2 plus energy intake; Model 2c: Model 2 without adjustment for prevalent diseases which may potentially lie on the causal pathway (adjusted for age, sex, BMI, smoking status, physical activity, alcohol intake, and, social economic status (income) only); Model 2d: Model 2 plus medication use (statins, antihypertensive medication, insulin and aspirin). *Analysis run excluding n=5 492 participants with comorbidities at baseline.</small>		Total flavonoid intake quintiles					Q1	Q2	Q3	Q4	Q5	Model 2	ref.	0.88 (0.85, 0.91)	0.83 (0.80, 0.86)	0.83 (0.80, 0.87)	0.83 (0.80, 0.87)	Model 2b	ref.	0.88 (0.85, 0.91)	0.83 (0.80, 0.86)	0.83 (0.80, 0.87)	0.83 (0.79, 0.87)	Model 2c	ref.	0.88 (0.85, 0.90)	0.82 (0.79, 0.85)	0.82 (0.79, 0.86)	0.83 (0.79, 0.87)	Model 2d	ref.	0.88 (0.86, 0.91)	0.83 (0.80, 0.86)	0.83 (0.80, 0.87)	0.83 (0.80, 0.87)	Model 2*	ref.	0.88 (0.85, 0.91)	0.82 (0.79, 0.86)	0.82 (0.78, 0.86)	0.82 (0.78, 0.86)	8
	Total flavonoid intake quintiles																																										
	Q1	Q2	Q3	Q4	Q5																																						
Model 2	ref.	0.88 (0.85, 0.91)	0.83 (0.80, 0.86)	0.83 (0.80, 0.87)	0.83 (0.80, 0.87)																																						
Model 2b	ref.	0.88 (0.85, 0.91)	0.83 (0.80, 0.86)	0.83 (0.80, 0.87)	0.83 (0.79, 0.87)																																						
Model 2c	ref.	0.88 (0.85, 0.90)	0.82 (0.79, 0.85)	0.82 (0.79, 0.86)	0.83 (0.79, 0.87)																																						
Model 2d	ref.	0.88 (0.86, 0.91)	0.83 (0.80, 0.86)	0.83 (0.80, 0.87)	0.83 (0.80, 0.87)																																						
Model 2*	ref.	0.88 (0.85, 0.91)	0.82 (0.79, 0.86)	0.82 (0.78, 0.86)	0.82 (0.78, 0.86)																																						
7	It should be noted that adding dietary variables to the model did attenuate the results somewhat.																																										
	We agree to elaborate on this finding and have added the following sentence to the manuscript: “In general, adjustments for potential lifestyle and dietary confounders (model 3) slightly attenuated the association and HRs for all flavonoid subclasses were not substantially lower beyond quintile 3”.	6																																									
8	On p. 12, although supplementary figure 5 does show a trend of greater reductions in overall and cancer mortality at higher levels of smoking or alcohol intake, this is not so clear for cardiovascular mortality																																										
	We have updated the results section of the manuscript to acknowledge this lack of a clear association for CVD-related mortality as follows: “Evidence of a dose-response association can be seen in Supplementary Figure 5, where the association between total flavonoid intake and both all-cause and cancer-	8																																									

	related mortality tended to stronger at higher smoking intensity (assessed by pack-years) and alcohol intake (g/d). Similar, although less clear, interactions between flavonoid intake and both smoking intensity and alcohol intake were seen for CVD-related mortality”.																														
9	The discussion focuses on improving flavonoid intake, but how might this differ from current health promotion efforts to increase vegetable and fruit intake? It would be good to see how an analysis of vegetable and fruit intake compares to the analyses of flavonoids with respect to mortality. Is there a benefit to making recommendations specifically around flavonoids rather than vegetables and fruit which may provide multiple beneficial components (fibre and micronutrients in addition to flavonoids)?																														
	In order to address this question, we have re-run the main analysis, stratified by total fruit and vegetable intake tertiles and have included this as a new Supplementary Table 3, as shown below. The association between total flavonoid intake and all-cause mortality remains amongst participants in the highest tertile of total fruit and vegetable intake. This is evidence that there may be further benefit to recommending a diet rich in flavonoids, above and beyond a diet rich in fruits and vegetables. This is reasonable as the main sources of flavonoids in this cohort were likely tea, wine, chocolate, apples and pears and, of these, only apples and pears are a fruit/vegetable. It is possible to have a low fruit and vegetable but high flavonoid diet and vice versa. We have added the following to the manuscript: Methods (page 17): “To determine whether the associations between flavonoid intake and all-cause mortality were independent of fruit and vegetable consumption, we stratified the analysis by tertiles of total fruit and vegetable intake”. Results (page 8): “When stratifying by tertiles of total fruit and vegetable intake, the association between total flavonoid intake and all-cause mortality remained amongst participants in the highest tertile (Supplementary Table 3)”. Discussion (page 9): “The possibility of flavonoids being a marker of other unobserved and potentially protective dietary factors cannot be discounted, although, in this study the associations between total flavonoid intake and mortality rates remained even after adjusting for other major indicators of a healthy diet as well as within the highest tertile of total fruit and vegetable intake”. Supplemental Table 3. Hazard ratios of all-cause mortality by quintiles of total flavonoid intake stratified by fruit and vegetable consumption    Total fruit and vegetable intake tertiles Total flavonoid intake quintiles   Q1 Q2 Q3 Q4 Q5     1 ref. 0.93 (0.86, 0.99) 0.89 (0.82, 0.96) 0.87 (0.79, 0.96) 0.88 (0.80, 0.97)   2 ref. 0.84 (0.77, 0.92) 0.87 (0.79, 0.96) 0.81 (0.73, 0.89) 0.79 (0.71, 0.87)   3 ref. 0.85 (0.72, 1.00) 0.86 (0.73, 1.00) 0.85 (0.73, 1.00) 0.91 (0.78, 1.06)    <small>Hazard ratios (95% CI) for 23-year all-cause mortality obtained from restricted cubic splines based on Cox proportional hazards models and stratified by tertiles of total fruit and vegetable intake. Analyses are adjusted for age, sex, BMI, smoking status, physical activity, alcohol intake, hypertension, hypercholesterolemia, social economic status (income), diabetes and prevalent disease.</small>	Total fruit and vegetable intake tertiles	Total flavonoid intake quintiles					Q1	Q2	Q3	Q4	Q5	1	ref.	0.93 (0.86, 0.99)	0.89 (0.82, 0.96)	0.87 (0.79, 0.96)	0.88 (0.80, 0.97)	2	ref.	0.84 (0.77, 0.92)	0.87 (0.79, 0.96)	0.81 (0.73, 0.89)	0.79 (0.71, 0.87)	3	ref.	0.85 (0.72, 1.00)	0.86 (0.73, 1.00)	0.85 (0.73, 1.00)	0.91 (0.78, 1.06)	8, 9 & 17
Total fruit and vegetable intake tertiles	Total flavonoid intake quintiles																														
	Q1	Q2	Q3	Q4	Q5																										
1	ref.	0.93 (0.86, 0.99)	0.89 (0.82, 0.96)	0.87 (0.79, 0.96)	0.88 (0.80, 0.97)																										
2	ref.	0.84 (0.77, 0.92)	0.87 (0.79, 0.96)	0.81 (0.73, 0.89)	0.79 (0.71, 0.87)																										
3	ref.	0.85 (0.72, 1.00)	0.86 (0.73, 1.00)	0.85 (0.73, 1.00)	0.91 (0.78, 1.06)																										
10	One of the novel features of the study is the examination of the subclasses of																														

	flavonoids. However, the estimated effect sizes for the subclasses appeared to be all essentially the same. Is this what would be expected biologically? It would be good to see a bit more discussion of the subclass results.	
	As the associations between flavonoid subclasses and mortality outcomes were investigated on a relative scale, their effect sizes cannot be compared. Participants in quintile 1 for flavonol intake may have differed from participants in quintile 1 for anthocyanin intake. In order to compare effect sizes between subclasses, analyses would need to be conducted on an absolute scale. We did not investigate the associations of subclass and mortality on an absolute scale (i.e. a Kaplan-Meier plot), due to the inability to adjust for confounding in such analysis. For this reason, we have refrained from discussing the effect sizes of the flavonoid subclasses. Due to differences in bioavailability and bioactivity, we would expect that the effect of flavonoid subclasses to be varying. However, this needs to be investigated on an absolute scale – something we are currently investigating.	
11	It is not clear why residual confounding is unlikely.	
	Residual confounding is unlikely as healthy lifestyle factors tend to cluster. This could be seen when we adjusted for potential confounders – as each additional confounder was added to the model, the change in risk estimates was attenuated. When adding other potential diet and lifestyle confounders to model 3, the hazard ratio did not change. However, upon reflection, we agree that we cannot determine the likeliness of residual confounding as there could potentially be another factor in a different cluster that could confound our results. As such we have removed that sentence from the manuscript.	9
12	The concern about having measurements at baseline only is not just that diet intake might change (and attenuate results), but also that potential confounders might change over time. For example, physical activity trajectories might differ in a manner associated with flavonoid intake and not be adequately captured by a baseline physical activity measure. It can be quite complex to tease apart multiple correlated factors and this complexity should at least be acknowledged.	
	We agree with the reviewer and have acknowledged this in the limitations section of the discussion by adding in the following sentence: “Furthermore, potential confounders were only assessed at baseline; it is unclear how changes in the trajectories of these confounders may have impacted upon the observed associations”.	9
13	The suggestion to develop high flavonoid content foods assumes causality, which may be premature. Also, caution is needed to ensure no harms are caused by radically increasing intake of any food component. Consider the controversies around folate fortification.	
	We thank the reviewer for pointing this out and have removed the sentence from the manuscript.	10 – 11
14	In table 1, since the proportion of women varies by quintile, the HRT distribution should be given for women only. Also in this table, it is not clear why aspirin is listed separately from NSAIDs since aspirin is an NSAID.	

	We thank the reviewer for pointing this out – we have updated Table 1 so that the HRT distribution is presented for women only, as suggested. We chose to present the prevalence of aspirin use separately to that of NSAID use as aspirin is used as secondary prevention for cardiovascular disease and has different medical indications to those of NSAIDs.	28
15	It is surprising that those in the lowest quintile of flavonoid intake have both a higher BMI (measured) and a lower energy intake (self-report from FFQ). Are there any concerns about differences in the accuracy of the FFQ across quintiles?	
	We have no concerns regarding the accuracy of the FFQ across the flavonoid quintiles. Participants in quintile 5 have a higher intake of fruits and vegetables, which contributes to the higher energy intake observed. They also have higher levels of physical activity so it is reasonable that they have a higher total energy consumption and lower BMI than those participants in quintile 1.	
16	In figure 4 and supplementary figures 3 and 4, it is confusing and a bit misleading that for smoking and alcohol the healthier group is represented by a solid line and the less healthy group by a dotted line, but this situation is switched for BMI. I suggest that the BMI groups be switched with respect to group representation in the figures for consistency and ease of interpretation.	
	We thank the reviewer for pointing this out. We have modified the figures as suggested.	
17	One thing that would make the paper and conclusions stronger would be the inclusion of a negative control analysis. This would be an analysis of total flavonoid intake with a cause of mortality not expected to be associated with flavonoids, such as accidents and injuries. If there is little suggestion of a relationship then it strengthens the conclusion of flavonoids contributing to reduced mortality from CVD and cancer. However, if similar estimated effects are observed it suggests that results may be driven at least in part by confounding or other bias.	
	We thank the reviewer for this suggestion and agree that no association between flavonoid intake and a falsification endpoint would strengthen the conclusion. Upon careful consideration, we decided that an ICD code for a foreign object or burn is an appropriate falsification endpoint as it should not be related to a disease state that could, in some way, potentially be influenced by a diet rich in flavonoids. We have included this in the manuscript as follows: Methods (page 17): “In order to assess the likelihood of confounding we used a falsification endpoint which we considered unlikely to be causally affected by flavonoid intake; any emergency, inpatient, or outpatient visit for a burn or foreign object (Supplementary Table 6)”. Results (page 8): “There was no association between total flavonoid intake and our falsification endpoint; a burn or foreign object [n=3 020; HR (95%CI) for Q5 vs Q1: 1.01 (0.92, 1.12); Supplementary Table 4”.	8, 9 & 17

Discussion (page 9):

“our findings are strengthened with a falsification endpoint analysis, which showed no association with total flavonoid intake”.

Supplementary Table 4. Hazard ratios of burns and foreign objects by quintiles of total flavonoid intake

	Total flavonoid intake quintiles				
	Q1	Q2	Q3	Q4	Q5
Model 2	ref.	0.95 (0.89, 1.02)	0.94 (0.86, 1.03)	0.97 (0.88, 1.07)	1.01 (0.92, 1.12)

Hazard ratios (95% CI) for any emergency, inpatient, or outpatient visit for a burn or foreign object (n=3 020) during 23 years of follow-up, obtained from restricted cubic splines based on Cox proportional hazards models. Analyses are adjusted for age, sex, BMI, smoking status, physical activity, alcohol intake, hypertension, hypercholesterolemia, social economic status (income), diabetes and prevalent disease.

REVIEWERS' COMMENTS:

Reviewer #1 (Remarks to the Author):

No further comments

Reviewer #2 (Remarks to the Author):

The authors have done a thorough job responding to the issues and I think that the changes generally make the paper clearer and stronger. I still have a couple of clarification issues.

1. In the response to point #9 regarding fruit and vegetable intake, the authors have included an analysis of flavonoids by tertile of fruit and vegetable intake, which is helpful. On page 8 it says that the relationship "remained strong amongst participants in the highest tertile". However, the results in supplemental table 3 show quite weak evidence for an inverse relationship between flavonoids and all-cause mortality in the highest tertile, tertile 3. The interpretation needs to be modified a bit, unless there is an error in the table.

2. The issue in point #14 may not have been sufficiently clear, but, since NSAIDs (cox1 and/or cox-2 inhibitors) as a group includes aspirin, if what is meant here is NSAIDs excluding aspirin, then it should be labelled correctly in the table.

The authors would like to thank the reviewers for taking the time to read through the updated manuscript.

Below is given a point-by-point response to the reviewer’s comments highlighted in bold and our answers are formatted in italic.

All indicated page numbers in this document refer to the manuscript file with highlighted (tracked) changes.

Reviewer 2

Qu	Reviewer comment/Author response	Page number
1	In the response to point #9 regarding fruit and vegetable intake, the authors have included and analysis of flavonoids by tertile of fruit and vegetable intake, which is helpful. On page 8 it says that the relationship "remained strong amongst participants in the highest tertile". However, the results in supplemental table 3 show quite weak evidence for an inverse relationship between flavonoids and all-cause mortality in the highest tertile, tertile 3. The interpretation needs to be modified a bit, unless there is an error in the table.	
	As suggested, we have modified the sentence accordingly: “When stratifying by tertiles of total fruit and vegetable intake, evidence of an association between total flavonoid intake and all-cause mortality remained amongst participants in the highest intake tertile”.	8
2	The issue in point #14 may not have been sufficiently clear, but, since NSAIDs (cox1 and/or cox-2 inhibitors) as a group includes aspirin, if what is meant here is NSAIDs excluding aspirin, then it should be labelled correctly in the table.	
	We have updated Table 1 so that NSAIDS are now labelled as “NSAIDS excluding aspirin”.	29